



# Different drought types and the spatial variability in their hazard, impact, and propagation characteristics

Erik Tijdeman[1], Veit Blauhut[2], Michael Stoelzle[2], Lucas Menzel[1], and Kerstin Stahl[2]

[1] Professorship of Hydrology and Climatology, Institute of Geography, Heidelberg University, Heidelberg, Germany
[2] Faculty of Environment and Natural Resources, University of Freiburg, Freiburg, Germany

*Correspondence to*: michael.stoelzle@hydro.uni-freiburg.de

**Abstract**

Droughts often have a severe impact on environment, society, and economy. Only a multifaceted assessment of such droughts and their impacts can provide insights in the variables and scales that are relevant for drought management. Motivated by this aim, we compared hazard and propagation characteristics as well as impacts of major droughts between 1990-2019 in Southwestern Germany. We bring together high-resolution datasets of air temperature, precipitation, soil moisture simulations, streamflow and groundwater level observations, as well as text-based information on drought impacts. Various drought characteristics were derived from the hydrometeorological and drought impact time series and compared across variables and spatial scales. Results revealed different drought types sharing similar hazard and impact characteristics. The most severe drought type identified is an intense multi-seasonal drought type peaking in summer, i.e. the events in 2003, 2015 and 2018. This drought type appeared in all domains of the hydrological cycle and coincided with high air temperatures, causing a high number and variability of drought impacts. The regional average drought signals of this drought type exhibit typical drought propagation characteristics such as a time lag between meteorological and hydrological drought, whereas propagation characteristics of local drought signals are variable in space. This spatial variability in drought hazard increased when droughts propagated through the hydrological cycle, causing distinct differences among variables, and regional average and local drought information. Accordingly, single variable or regional average drought information is considered to be not sufficient to fully explain the variety of drought impacts that occurred. In addition to large-scale drought monitoring, drought management needs to consider local drought information from different hydrometeorological variables and could be type based.

## 1 Introduction

The Central and Northern European drought and heatwave of 2018 revealed once more the large spatial-temporal footprint and severe impacts of this natural hazard (e.g. Bakke et al., 2020; Brunner et al., 2019; Schuldt et al., 2020). Similar or worse episodes are expected to occur more often in the future, given the increasing atmospheric water demand and human pressure



on fresh-water resources (e.g. Samaniego et al., 2018; Wanders and Wada, 2015). This prospect raises the importance of short- and long-term drought management to better cope with both ongoing drought as well as to be better prepared for future drought episodes (Wilhite et al., 2019). Good planning for drought requires information about the different components of drought risk: hazard, impacts, exposure, and vulnerability. The analysis of past droughts at different scales can provide this information.

On the one hand, locally relevant drought management benefits from detailed information, which considers different hydrometeorological variables and drought related impacts and their spatiotemporal variability (e.g., Van Lanen et al., 2016). On the other hand, higher-level administrative decision-making often requires drought information in a more generalized form, e.g., indexed information aggregated to administrative regions indicating whether there is drought or not. Generalizing drought information simplifies its interpretation but may come at the cost of a loss of information, as the hazard and its impacts may

be highly variable in space and time (Stahl et al., 2016).

The multifaceted nature of drought results in the impracticability of a unique drought definition (Lloyd-Hughes, 2014). From a hazard perspective, drought is often defined as a below normal hydrometeorological anomaly, where the normal depends on space, time, and the variable of interest (e.g., Tallaksen and Van Lanen, 2004). Accordingly, a wealth of drought hazard indices has been developed to express the flux or state of a certain domain of the hydrological cycle, or the combined states and fluxes

of multiple domains, as anomaly (e.g. Hao and Singh, 2015; Zargar et al., 2011). From an impact perspective, below normal anomalies only become droughts when they have the potential to cause drought impacts, i.e., "negative consequences of drought for environment, society or economy" (Blauhut et al., 2015).

The above-described definitions of drought form the basis of many drought-related studies, which often focus on a specific type of drought such as meteorological, agricultural, hydrological, or socio-economic drought (Wilhite and Glantz 1985).

Laaha et al. (2017) argue that our understanding of drought would benefit from a more holistic study of drought phenomena, because specific drought impacts relate to droughts in certain domains of the hydrological cycle. In addition, the co-occurrence of drought in different domains of the hydrological cycle may worsen drought impacts. For example, agricultural drought impacts caused by low soil moisture can be aggravated by co-occurring streamflow droughts that limit or prohibit withdrawals of surface water for irrigation. A holistic view on drought and its impacts might further benefit the consideration of compound

hazards preceding or co-occurring with drought, e.g., heat waves, as these can further worsen drought impacts (Zscheischler et al. 2020). Finally, a temporal clustering of drought years might be worth the consideration. A second drought year in a row, as observed for meteorological drought for Central and Northern Europe in 2018-2019 as well as for various hydrological droughts in the UK (resp. Hari et al., 2020; Kendon et al., 2013), prevents recovery and might have an even larger impact on already weakened systems.

The connection of different drought types through so-called "drought propagation" (e.g., Changnon, 1987; Van Loon, 2015) further justifies the need for a holistic drought assessment. Drought propagation is a well-established concept on the catchment scale (e.g. Van Loon, 2015). Rainfall deficits enhanced by meteorological conditions that favor high evapotranspiration propagate to deficits in root zone soil moisture followed by recession and ultimately deficits in river flow and groundwater. Catchment scale drought propagation exhibits various typical characteristics, including the order of appearance of drought in



different domains of the hydrometeorological cycle and associated time-lag as well as the attenuation of the propagating hydrometeorological signal and associated lengthening. Regional scale drought propagation is a less well-established concept. On this regional scale, drought propagation may not happen uniformly, as the propagating drought signal is affected by meteorological variability and further modified by geological differences in surface and sub-surface characteristics. Differences in meteorological conditions are driven by both larger- and smaller-scale atmospheric processes. On the larger

(continental) scale, atmospheric circulation patterns and teleconnections play an important role, e.g., by blocking wet weather systems (e.g., Ionita et al., 2017; Toreti et al., 2019). On the smaller regional scale, differences in meteorological conditions appear due to the occurrence of local rainstorms or differences in topography and land cover that affect energy balance terms and hence air temperature, evapotranspiration, and meso-scale wind systems. Differences in subsurface soil and geological characteristics may also exert a strong control on regional drought propagation, as these affect the total amount of storage and

thereby influence how well meteorological dry spells can be buffered (e.g., Barker et al., 2016; Bloomfield and Marchant, 2013; Heudorfer et al., 2019; Stoelzle et al., 2014).

The variable occurrence and severity of drought in space, time, and among different domains of the hydrological cycle may be a major precursor of when and where different types of drought impacts occur. However, drought impact occurrence is not solely related to the hazard as drought impacts are "symptoms of vulnerability" (Knutson et al, 1998). Thus, an indicated

drought hazard might not necessarily lead to an impact, as the exposed system also has to be vulnerable to the hazard. For example, certain drought impacts might not occur because of appropriate mitigation measures in place. In addition, the considered spatially aggregated or single variable drought hazard information might not be representative for a specific type of local drought impacts.

It is the question whether a holistic assessment of droughts enhances the understanding of the type and amount of drought

impacts that occur during certain types of drought events, as this could highlight the need for multivariate drought type specific management. It is further the question whether concepts of catchments scale drought propagation also exist on the regional scale, as these concepts could serve as guiding principles for regional drought management. For both these questions, spatial scale, i.e., whether local or regional aggregated drought information is investigated, may play an important role. What is the unique value of different drought information sources at different scales? And is there a benefit of using local over spatially

aggregated drought information in a regional drought management context? These questions have not yet been systematically explored as often data is lacking for such comprehensive analyses. To fill this gap, this study aims to shed more light on regional drought (propagation) characteristics with a multivariate view on recent droughts to enhance the understanding of drought impacts. Focusing on Southwestern Germany, we aim to:

1.  identify drought episodes for the period 1990-2019 and investigate if these episodes can be grouped into different

types with similar hazard characteristics and impacts,

2.  evaluate the propagation of drought over the region for the most prominent (highest impact) type's drought episodes, and

3.  assess the agreement among drought hazard and impact information across variables and scales.





## 2 Data and methods

**2.1 Study area**

Baden-Württemberg is the most Southwestern federal state of Germany (Fig. 1). According to the Nomenclature of Units for Territorial Statistics (NUTS) of the European Union, Baden-Württemberg is a NUTS-1 region, which is separated into four NUTS-2 regions: Stuttgart, Karlsruhe, Freiburg and Tübingen. In this case, these NUTS regions also have their own governmental water authorities. Hydrologically important, the river Rhine flows along the southern and western border of the

state and provides an important waterway for navigation. The region encompasses both flat and lowland areas such as the Rhine Valley in the West as well as more mountainous regions such as the Black Forest and the Swabian Alb. The topography of the region affects both precipitation (annual average sum between 600 and >2000 mm year$^{-1}$) and air temperature (annual average between 4.5 and 11.6 °C). The landscapes of Baden-Württemberg are diverse, with agricultural (43%) and urban (7%) areas mostly located in the lower elevated regions and forested (38%) and meadow areas (10%) in the higher elevated regions.

Equally diverse are the region's lithological and geological characteristics. A variety of soils with different root-zone depths and water holding capacities exist, ranging from thick loess layers to shallow Leptosols. Below these soils, the geology varies from metamorphic rock to porous limestone and unconsolidated rock, resulting in a hydrogeology with different aquifer types that have different storage capacities (Stoelzle et al., 2015). Aquifers also vary in size, with some larger porous aquifers in e.g., the Rhine Valley and smaller valley fill aquifers among fractured sedimentary or crystalline bedrock in e.g., the Black Forest.

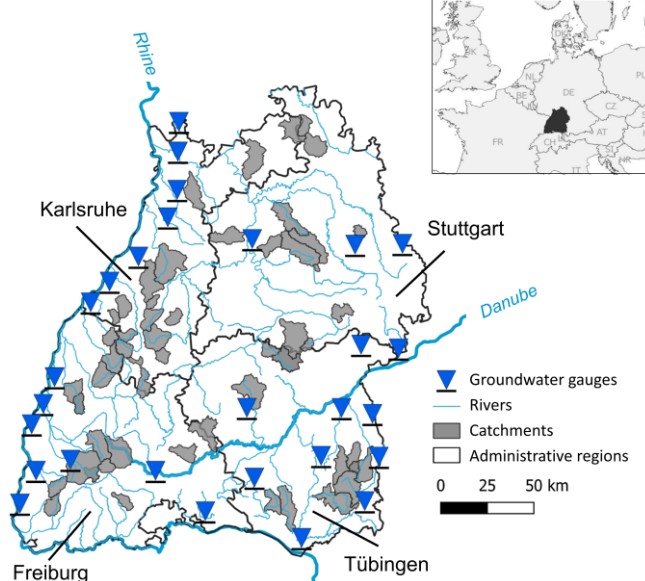


**Figure 1. Data basis for the study in the federal state of Baden-Württemberg (NUTS-1) with its four different administrative regions (NUTS-2).**



## 2.2 Data and temporal aggregation

The study assembled multiple variables for the region: air temperature, precipitation, simulated soil moisture and observed
streamflow and groundwater levels. These daily hydrometeorological observations and simulations for the period 1990-2019
stem from different sources. Air temperature data come from own climate station interpolations over a 1-km resolution grid
covering Baden-Württemberg (Tijdeman and Menzel 2021a). Precipitation data stem from the gridded REGNIE data product
(Rauthe et al., 2013) and was sourced from the Climate Data Center of the German Weather Services (DWD, 2020). Gridded
simulated soil moisture data over a 1-km resolution grid were derived using the TRAIN model (described in Tijdeman and
Menzel 2021a) and were obtained from Tijdeman and Menzel (2021b). Daily streamflow observations at gauging stations of
54 catchments with flow unaffected by major human disturbances at the timescale relevant for our analyses were provided by
the Ministry of Environment of Baden-Württemberg (LUBW). Groundwater level observations of 43 wells were obtained from
the Environmental Data and Maps archive of the LUBW (UDO, 2020). We only considered those groundwater wells currently
in use in the reference network of the groundwater storage assessment application of the LUBW (LUBW, 2020), given their
relevance for drought monitoring. This does not mean that all groundwater level time series are completely free of human
influences. Nevertheless, Baden-Wurttemberg is a water rich country, and visual inspection of the groundwater level time
series did not show any sharp changes associated with changes in (nearby) abstractions.

All above-described variables are continuous daily observations, except for groundwater level observations, which are
available at (ir)regular intervals, e.g., at certain days of the week. To obtain a common temporal resolution, we aggregated
daily hydrometeorological data to a monthly resolution by taking the mean of the daily observations or simulations. This
resulted in monthly time series of precipitation (P1), air temperature (T1) simulated soil moisture (SM) streamflow (Q) and
groundwater levels (GW) for either each grid cell (P1, T1, SM), catchment (Q), or well (GW). Precipitation was further
aggregated to a seasonal and annual resolution (resp. P3 and P12), one value for each calendar month, to account for both
short- and long- term meteorological deficits relevant for e.g., hydrological systems with resp. a low- and high- buffering
capacity of meteorological deficits. Air temperature was also aggregated to an annual resolution (T12), one value for each
year, to depict the general changes in climate. Daily time series of precipitation, air temperature and simulated soil moisture
did not contain any missing values, whereas daily time series of streamflow and groundwater level observations contained
occasional gaps. In the case of gaps in daily streamflow, a certain month was set to missing whenever five or more days in that
month did not have an observation. For groundwater, we used less stringent missing data requirements, given that groundwater
levels were not available at a continuous daily resolution. GW in a certain month was set to "missing" when there were no
observations in that month. In the end, we only selected the monthly Q and GW level time series with less than 10% of
"missing" months, respectively all of the 54 streamflow time series and 28 out of 43 of the groundwater level time series (Fig.
1).

Drought impact information stems from the European Drought Impact report Inventory (EDII; Stahl et al., 2016); a database
of textual drought impact information from different sources, e.g., newspaper articles, governmental reports etc. These reports





are manually collected, temporally and spatially referenced, and classified to one of 15 impact categories (Blauhut et al., 2015). The timestamp of an entry indicates at a minimum the year of occurrence, but where available also information on the beginning and in some cases the end date of a reported impact. For this study, we considered impacts for the period 2000-2019 (n = 792), given the more limited availability of impact information prior to this date. The EDII has recently been updated for

the greater alpine region (Stephan et al., 2021). For the study region of Baden-Württemberg this dataset was also supplemented with additional impact information gathered from questionnaires of a survey of the hydropower sector (Siebert et al. 2021; until 2017) and a survey of the public water supply sector (Blauhut et al. 2020; until 2018). Categorical impact information was grouped by start year, season, where available start month (n = 359), and NUTS-1 and NUTS-2 region. The grouping over NUTS regions was used as these regions coincide with administrative boundaries relevant for decision-making according to

the state's Water Act.

**2.3 Drought hazard information at different spatial scales**

We derived drought hazard information from all meteorological and hydrological variables at three spatial scales: local, NUTS-2 average, and NUTS-1 average. Local drought information was derived for each individual grid cell, catchment or well by transferring their time series to anomaly space (percentiles; $p$) using Weibull plotting positions (Weibull 1939, eq. 1).


$$p_{V,u} = \frac{\text{Rank}(V_u)}{(n+1)} \qquad \text{(eq. 1)}$$

Where $V$ is the variable of interest, $n$ the sample size (in this study: $n = 30$) and u the location identifier referring to either a grid cell, catchment, or groundwater well. The rank of $V$ in a specific year and month is relative to historical observations in

that month, e.g., P1 for August 2018 compared to P1 for all other Augusts. To derive anomalies from air temperature time series (T1, T12), we used an inverse ranking to make the percentile classification of above normal air temperature comparable with below normal hydrometeorological conditions. The derived percentiles express the historical non-exceedance frequencies of different variables for the considered reference period 1990-2019 (Tijdeman et al., 2020). This reference period differs from the standard reference period of the WMO (e.g., 1961-1990). However, using the period 1990-2019 as reference was preferred

given the lower availability of continuous observations for the earlier period.

For the NUTS-1 region as a whole and for the four NUTS-2 sub-regions, regional average drought hazard information was derived by ranking the average of all available local percentile data within the region; for a given hydrometeorological or hydrological variable, the regional average $p$ is then (eq. 2).


$$p_{V,\text{NUTS}} = \frac{\text{Rank}(\overline{p_{V,\text{NUTS}}})}{(n+1)} \qquad \text{(eq. 2)}$$

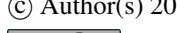


Where NUTS refers to either all grid cells, catchments or wells located within either the NUTS-1 or NUTS-2 region of interest. The rank of $\overline{P_{V,\text{NUTS}}}$ thus compares average local percentile values for a specific NUTS-1 or NUTS-2 region, variable, year, and month to average percentile values for the same NUTS-1 or NUTS-2 region, variable, and month in all other years.

The percentile time series of all hydrometeorological variables and scales were classified into five different groups: $p_V \leq 0.1$ (severe drought or much above normal air temperature), $0.1 < p_V \leq 0.25$ (moderate drought or above normal air temperature), $0.25 < p_V < 0.75$ (normal conditions), $0.75 \leq p_V < 0.9$ (moderately wet or below normal air temperatures) and, $p_V \geq 0.9$ (severely wet or much below normal air temperatures). For ease of notation, we move the location identifier outside of the subscript and use the variable abbreviation to refer to its percentiles where applicable, e.g., local P1 when referring to monthly precipitation

percentiles at individual locations (cells) or NUTS-1 average P1 when referring to regional average monthly precipitation percentiles.

**2.4 Typing hazard characteristics and impacts**

We characterized past drought episodes over the period of 1990-2019 and the impacts of recent events from 2000-2019. First, we identified drought episodes from the NUTS-1 average percentile time series of all hydrometeorological variables. These

episodes were classified into different types considering timing and length of the episode as well as the affected domains of the hydrological cycle. These aspects of drought episodes were considered as we hypothesize that they influence the type and amount of related drought impacts that occur.

In a next step, we quantified whether different types of drought episodes also differ in hazard and impact characteristics. We first divided the NUTS-1 average percentile time series ($p$) of all hydrometeorological variables ($v$) into drought ($p_V \leq 0.25$)

and non-drought events ($p_V > 0.25$). We then derived the duration ($D$, months) and severity ($S$, -) of each event $j$ (eq. 3 & 4, respectively):

$$D_{p_V,\text{total}}[\,j\,] = \sum_{t=1}^{D_{P_V}(j)} 1 \quad (\text{eq. 3})$$

$$S_{p_V,\text{total}}[\,j\,] = \sum_{t=1}^{D_{P_V}(j)} 0.25 - p_V(t) \quad (\text{eq. 4})$$


$S$ hence incorporates both the duration of the event and the deviation from the threshold but has no physical relation to water quantity. $S$ should be seen as a relative metric that only enables comparison among other variables' severity values. With regard to drought impact characteristics, we considered the number and categories that happened during the different types of drought episodes. The number of impacts gives an indication of the severity and perception of the drought events, whereas impact

categories provide insight in the (diversity of) affected sectors.



## 2.5 Regional drought propagation

We introduce regional drought propagation, i.e., the propagation of drought through the hydrological cycle in space and time over a larger region. Focusing on the most prominent and also highly impacted drought episodes, it was investigated whether regional drought propagation exhibited typical characteristics known from catchment scale drought propagation such as
ordering, time-lag, and lengthening. We first inspected whether these drought propagation characteristics were present in NUTS-1 average, NUTS-2 average, and local percentile time series. We then quantified the initiation time ($I$, months) and maximum duration ($M$, months) of each drought episode from both NUTS-1 average and all local percentile time series of all hydrometeorological variables ($p_V$), where:

-    $I_{p_V}$ is the initiation time, i.e., the time between an arbitrary set starting point (e.g. start of drought development) and
220            the first time the percentile time series reached drought ($p_V \leq 0.25$), and

-    $M_{p_V}$ is the maximum time (months) the percentile time series was continuously in drought ($p_V \leq 0.25$) during the
            drought episode.

From $I_{p_V}$ and $M_{p_V}$ of all local percentile time series of each variable, we derived the 5th and 95th quantile to differentiate between quick ($I_{p_V,Quick}$) and slow ($I_{p_V,Slow}$) developing drought conditions as well as short ($M_{p_V,Short}$) and prolonged
($M_{p_V,Long}$) local drought conditions.

The ordering and difference of $I_{p_V}$ among variables is indicative for resp. the ordering and time lag between drought events appearing in different domains of the hydrological cycle. An amplification in $M$ when drought events propagate through the hydrological cycle is indicative of lengthening. We hypothesize that the order of appearance, time lag, and lengthening of drought are generally visible in the NUTS-1 average and local drought signals such that:

-    $I_{P1} \leq I_{SM} \leq I_Q \leq I_{GW}$

-    $M_{P1} \leq M_{SM} \leq M_Q \leq M_{GW}$

However, we also speculate that regional differences in climate, soil, catchment, and aquifer characteristics can modify the order of appearance and lengthening of local drought conditions over a larger region such that:

-    $I_{SM,Slow} \geq I_{Q,Quick}$; $I_{Q,Slow} \geq I_{GW,Quick}$
-    $M_{SM,Long} \geq M_{Q,Short}$; $M_{Q,Long} \geq M_{GW,Short}$

## 2.6 Agreement among different drought information sources

The usefulness of different levels and sources of drought information was assessed based on their agreement ($A$; between 0 and 1). The lower $A$ among two different drought information sources, the higher the unique value of each individual source. For this analysis, one prominent drought episode was chosen. For this episode, we derived $A$ by comparing the fraction of
cases each individual drought ($p_V < 0.25$) information source (different variables and different scales) was in drought for all months in which:





- The NUTS-1 average percentile time series of P1, SM, Q or GW was in drought, or

- at least one drought impact started.

To gain more insight in the variability in hydrometeorological conditions beyond agreement metric *A*, we also derived the
percentile class distributions of the NUTS-1, NUTS-2, and local percentile time series for all months in which the NUTS-1
average percentile time series was in drought or all months in which at least one drought impact report started.

## 3 Results

### 3.1 Different drought episode types and their characteristics and impacts

NUTS-1 average percentile time series of different variables reveal several past drought episodes in the study region (Fig. 2).
Based on common characteristics, these episodes can be grouped into three distinct types and two combinations of them:

- intense multi-seasonal drought episodes peaking in summer as in 2003, 2015 and 2018 (Type I),
- long-term and less intense drought episodes in e.g., the early 1990s (Type II),
- short-term episodes as in e.g., 2011 & 2014 (Type III),
- Type I episodes that transition in Type II episodes as in e.g., 2004 & 2019 (referred to as Type I→II) and,
- Type III episodes that coincide with Type II episodes as in e.g., 1998 (referred to as Type III+II).


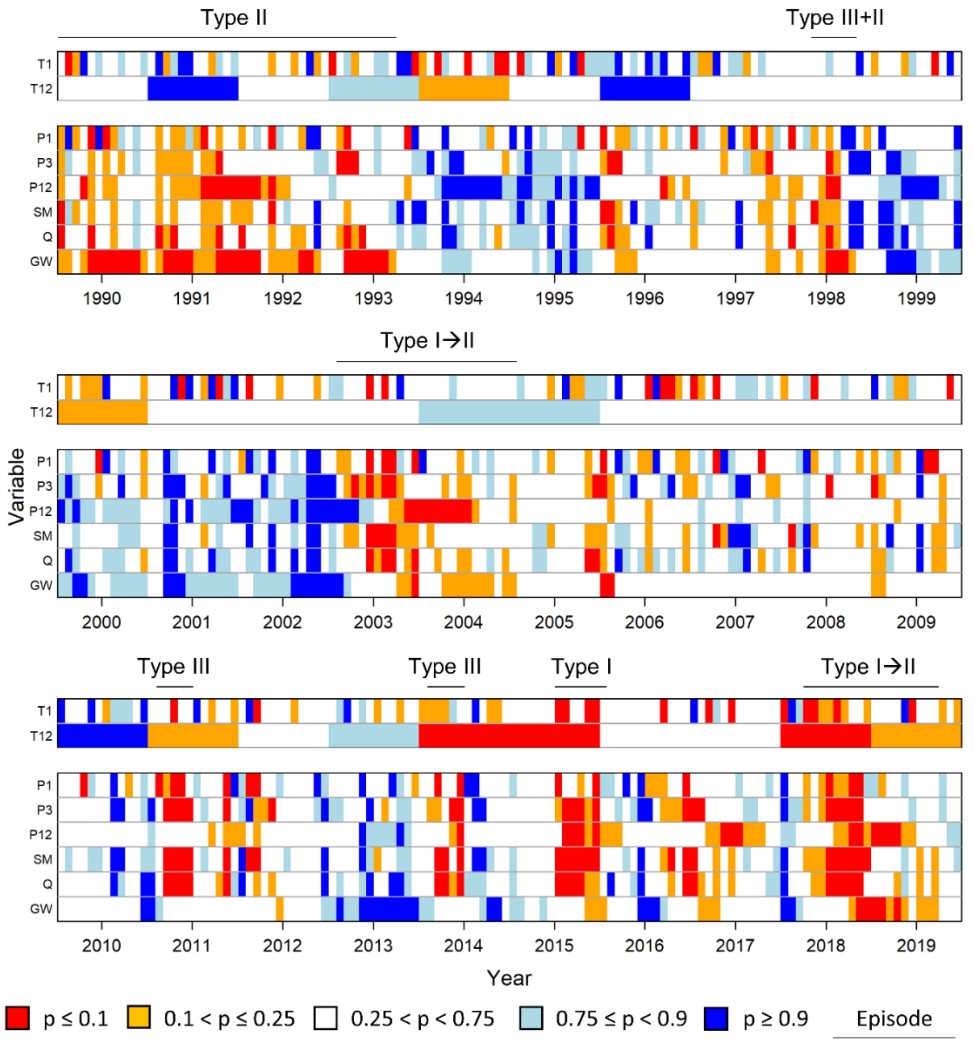

**Figure 2. Drought episode types identified from NUTS-1 average classified percentile (p) time series of monthly and annual air temperature (resp. T1 and T12), precipitation accumulated over different periods (P1, P3 and P12), simulated soil moisture (SM) and observed streamflow (Q) and groundwater levels (GW).**

Type I episodes showed persistent meteorological dry spells over the growing season according to NUTS-1 average P1 and

P3 that caused severe and prolonged deficits in NUTS-1 average SM and Q. Type II and Type I→II episodes were characterized

by long-term meteorological dryness as indicated by NUTS-1 average P12 and were associated with persistent below normal

NUTS-1 average GW, occasionally interrupted by some wetter months. The shorter Type III episodes caused below normal

NUTS-1 average SM and Q. However, the impact of these shorter dry spells on NUTS-1 average GW varied among the years

and depended on the initial conditions of the groundwater systems at the start of the drought. The Type III+II episode of 1998

was preceded by a period of long-term dryness and coincided with below normal NUTS-1 average GW, whereas the Type III

episodes of 2011 and 2014 happened after a relatively wet period and did not cause below normal NUTS-1 average GW.





The temperature setting in which these different types of drought occurred varied among drought years. There is a general increasing trend towards higher annual air temperatures and NUTS-1 average T12 of four out of the last six years (2014-2019) was warmer than average (relative to the period 1990-2019). Nevertheless, there can be a lot of within year monthly variability according to NUTS-1 average T1. Type I episodes all coincided with at least a few months of extremely high air temperatures. On the other hand, the Type II episode of the early 1990s happened in a relatively cold setting. Air temperature was below normal for the Type III+II episode that occurred in 1998, whereas the Type III episodes in 2011 and 2014 coincided with one or a few warmer months.

The duration ($D$) and severity ($S$) of past drought episodes derived from NUTS-1 average percentile time series vary among variables and drought types (Fig. 3). $D$ and $S$ of meteorological dry spells was generally low (Fig. 3a-b). A notable exception was the meteorological drought of 2018 with $D_{T1}$ and $D_{P1}$ of six months. For Type I droughts, $D$ and $S$ increased when droughts propagated from precipitation to soil moisture and streamflow (Fig. 3c-d). Such an amplification in drought characteristics was not visible for the other drought types. On the other hand, the $D_{GW}$ and $S_{GW}$ of type II episodes were exceptional (Fig. 3e). For this drought type, multiple shorter meteorological drought events coincided with prolonged periods of groundwater drought. $D_{GW}$ and $S_{GW}$ were also high after part of the Type I drought episodes (2004, 2018-2019).

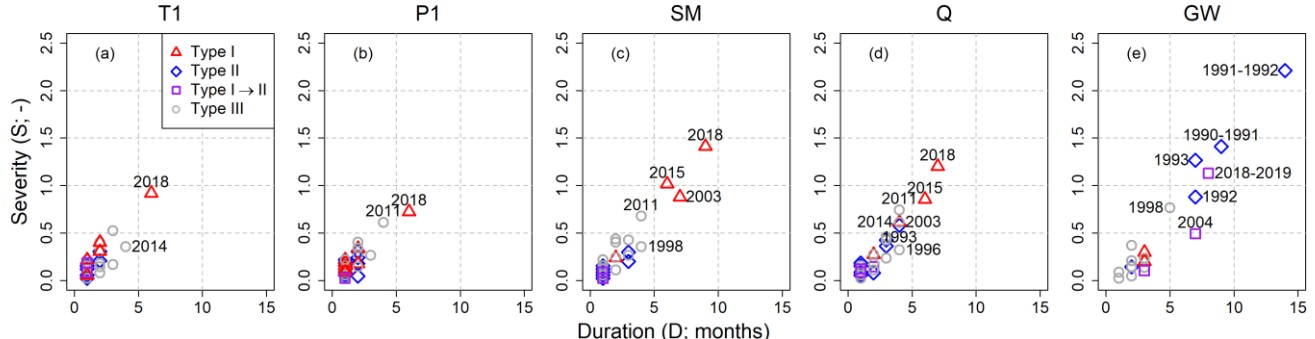

**Figure 3. Duration ($D$) and severity ($S$) of hydrometeorological drought events during different types of drought episodes derived from monthly NUTS-1 average percentile time series with (a) air temperature (T1) (b) precipitation (P1), (c) simulated soil moisture (SM), (d) streamflow (Q), and (e) groundwater (GW). Events with a duration ≥ 4 months are labeled.**

The total number of reported drought impacts varied among drought years as well as drought types (Fig. 4a). Most impacts were reported for the Type I episodes, lesser for the Type III episodes and least for the Type I→II episodes. The categorical composition of drought impacts revealed that the largest shares of impacts related to the energy and industry sector, agriculture and livestock farming, public water supply, and freshwater ecosystems (Fig. 4b). The most frequently reported impact type in the category of energy and industry was reduced hydropower production. Agricultural impacts include the reduction in harvest quantity and quality or the restriction in irrigation. Public water supply impacts are often related to water use restrictions or the need to allocate water from other sources, whereas freshwater ecosystem impacts e.g., related to fish die-off. In addition to the more commonly reported impacts, a large variety of other impacted sectors became visible, including forestry, water

quality, waterborne transportation, tourism and recreation, and drought related conflicts. The categorical distribution of reported impacts for Baden-Württemberg further reveals drought-type specific differences (Fig. 4b). In general, the highest diversity in impact categories was reported for the Type I drought episodes, whereas drought impacts of Type I→II and Type III episodes were less diverse. A comparison by years further shows event-specific differences. In 2015 and 2018, a relatively large share of impacts on agriculture were reported. The drought of 2011 was dominated by impacts attributed to hydrological drought, i.e., anomalies in streamflow and groundwater levels, such as energy and industry, waterborne transportation, and

public water supply. However, no impacts on forestry and agriculture were observed. In contrast, the year 2019 showed a large share of forestry related issues, especially related to the dieback of spruce and bark beetle infestations.

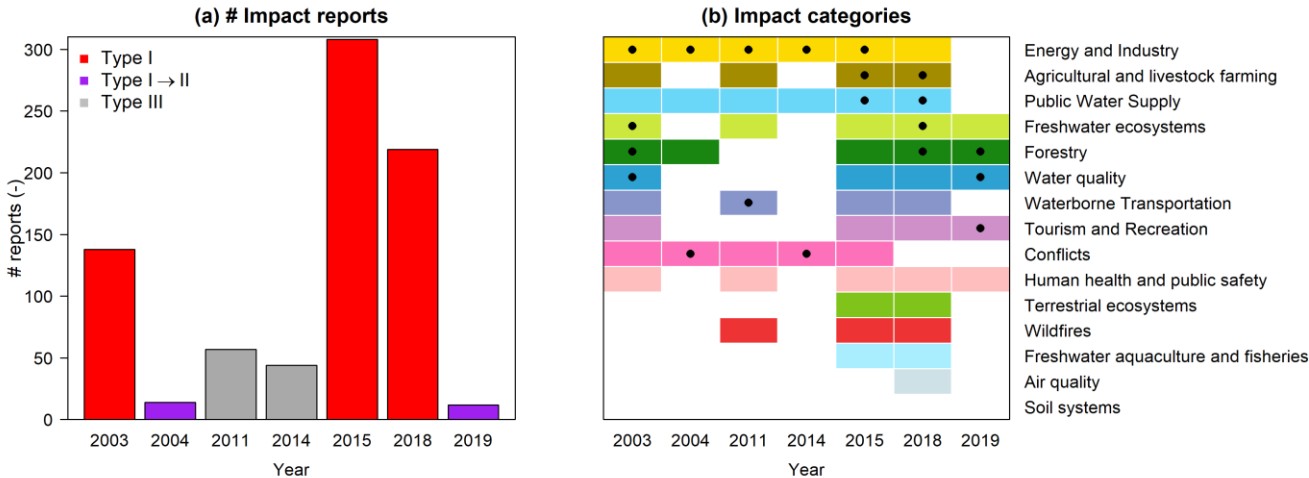

**Figure 4. Reported impacts of drought episodes from 2000 to 2019. (a) Annual number of impacts in the European Drought Impact Inventory by drought type, and (b) reported drought impact categories for different drought years, sorted descending by total number of reports. Black dots indicate impact categories that account for more than 10% of all impact reports in a specific year.**

## 3.2 Regional drought propagation characteristics of prominent Type I and Type I→II episodes

A similarity for the prominent Type I and Type I→II drought episodes is their common start from relatively wet initial conditions in winter (Fig. 5). From that point, several months with below normal P clustered together in prolonged periods of below normal SM and Q, and eventually reached a state of below normal GW. Another similarity is that local drought signals

become more variable in space when propagating through the hydrological cycle, as is shown by the increasing ranges in local drought conditions. A further similarity among prominent drought years is the occurrence of some relatively wet months in the following winter. In 2003 and 2018, these single wet months generally recovered drought for SM and Q, but only had a small impact on GW, which often stayed low or continued to decline. The wet January in 2004 was not sufficient for a full recovery of GW, which meant that part of the groundwater levels dropped again to below normal conditions that would persist

for the remainder of 2004 and the first part of 2005. The same was observed for the drought starting in 2018; some relatively wet winter months had little effect on most of the groundwater levels, which stayed below normal throughout the year 2019.




On the other hand, the drought of 2015 was followed by multiple wet winter and spring months, which alleviated drought conditions by the start of the summer of 2016.

**Figure 5. Spatiotemporal propagation of prominent droughts (Type I and Type I→II) according to (ranges in) NUTS-1, NUTS-2 and local percentile timeseries relative to the reference period 1990-2019 (Sect. 2.4). Dashed line indicates the moderate drought or above normal air temperature threshold.**

Various typical drought propagation characteristics, i.e., ordering, time-lag, and lengthening, can be recognized from the (ranges in) NUTS-1, NUTS-2, and local percentile time series (Fig. 5). The occurrence of these drought propagation characteristics is partly confirmed when comparing initiation time ($I_V$) and maximum duration ($M_V$) among the different variables (Fig. 6). For all three drought episodes, the hypothesized ordering and time lag is visible in NUTS-1 $I_V$ and local $I_{V,Quick}$ (Fig. 6a; i.e., $I_{P1} \leq I_{SM} \leq I_Q \leq I_{GW}$, e.g., for 2003: $2 \leq 3 \leq 6 \leq 10$ months). Local $I_{V,Slow}$ shows this expected ordering for P, SM and Q. However, not all local GW observations reached drought during the drought episodes of 2003 and 2015. The





lengthening of drought is visible when P1 droughts propagate to SM droughts ($M_{P1} \leq M_{SM}$). However, $M_Q$ is occasionally

lower than $M_{SM}$ (e.g., 2003) and $M_{GW}$ is occasionally lower than $M_Q$ (e.g., 2015; Fig. 6b).

The ranges in Figure 5 indicate that local drought conditions within a region according to the same variable may vary strongly

at one moment in time. The variation also suggests that the temporal sequencing of different drought types was not uniform

over a larger region. This is confirmed when comparing local $I_{V,Quick}$ and $I_{V,Slow}$ (diagonal arrows in Fig. 6a). For example, GW

of some responsive aquifers reached below normal conditions prior to Q of some less responsive catchments ($I_{Q,Slow} \geq I_{GW,Qucik}$

e.g. for 2003: $12 \geq 4$ months). This is also confirmed when comparing $M_{V,Short}$ and $M_{V,Long}$. For example, the longest time local

SM was continuously in drought exceeds the shortest time local Q was continuously in drought ($M_{SM,Long} \geq M_{Q,Short}$ e.g. for

2003: $11 \geq 2$ months).

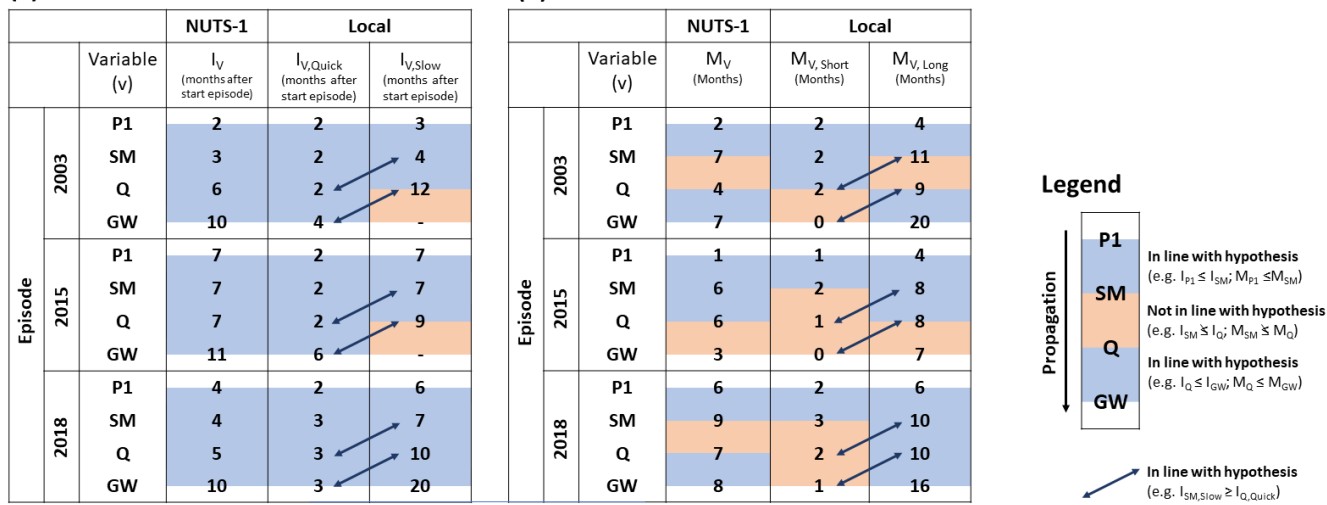

**Figure 6.** Typical NUTS-1 average and local drought propagation characteristics (a) initiation time ($I_V$, $I_{V,Quick}$ & $I_{V,Slow}$; in months
after the start of the indicated year), and (b) maximum duration ($M_V$, $M_{V,Short}$ & $M_{V,Long}$; months) derived from the and local
percentile time series of monthly precipitation (P1) simulated soil moisture (SM) streamflow (Q) and groundwater (GW) for the
three prominent drought episodes. Colors reflect whether drought propagation characteristics are in line with our hypothesis (Sect.
2.5).

**3.3 Agreement among drought information sources: the case of the drought of 2018-2019**

The variability in propagating drought signals affects the agreement between different drought information sources. This

agreement ($A$) among different drought information sources is indicative of the unique value of an individual source as

indicator of drought occurrence. In general, this agreement as well as the classified percentiles of different variables at different

scales, reveal that hydrometeorological conditions can be quite variable whenever NUTS-1 average conditions of one variable

indicated drought (Fig. 7). Strongest agreement was found between NUTS-1 and NUTS-2 or local percentile time series of the

same variable for P1, SM and Q (Fig. 7a-c). Whenever the NUTS-1 average percentile time series of P1, SM or Q were in

drought, NUTS-2 average and local percentile time series of these variables often showed drought conditions as well ($A = 75$-



90%). However, lesser agreement was observed between NUTS-1 average GW and NUTS-2 average and local GW ($A$ = 50-70%; Fig. 7d). Lesser agreement was also observed among NUTS-1 average and NUTS-2 average or local percentile time series of different variables. Whenever NUTS-1 average P1 was in drought, NUTS-1 average, NUTS-2 average and local

percentile time series of other variables were as well for 25-60% of cases (Fig. 7a).  SM and Q agree relatively well with each other ($A$ = 70-85%) but not so much with P1 and GW (A= 50-80%; Fig. 7b-c). NUTS-1 average GW often did not agree with percentile time series of other variables ($A$ = 30-50%; Fig. 7d). The agreement between impact start and drought hazard information reflects how well different drought hazard information sources can predict the start date of an impact. In general, it can be seen that hydrometeorological conditions can be quite variable at the start date of an impact (Fig. 7e). Strongest

agreement between impact start and drought hazard occurrence was found for SM and Q ($A$ = 60-75%). Lesser agreement was found between impact start and drought hazard occurrence according to P and GW ($A$ = 30-55%).

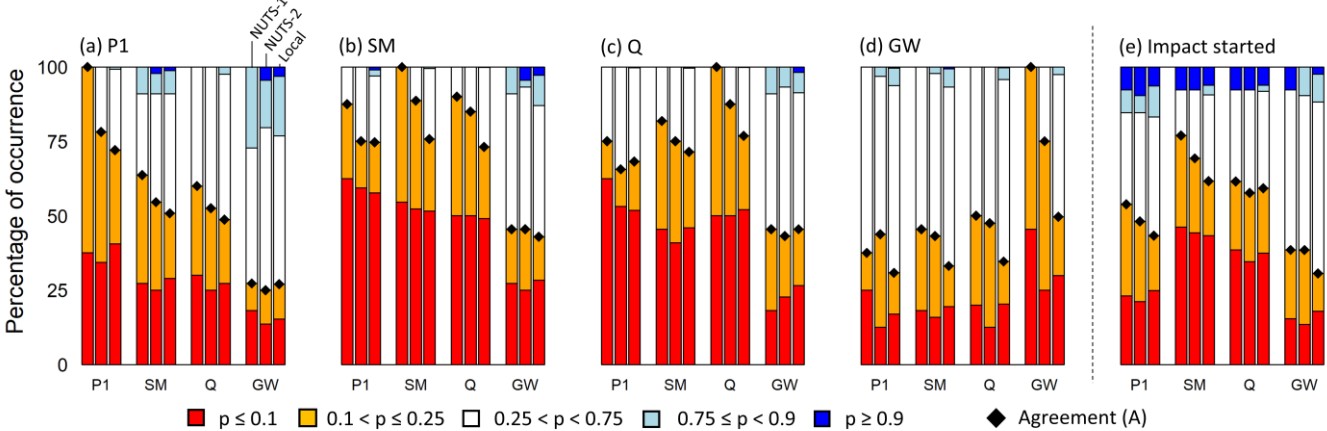

**Figure 7. Classified percentile ($p$) distributions of different variables (x-axis) at different scales (bar triplets: NUTS-1, NUTS-2 and local) during the drought episode of 2018-19 for all months with NUTS-1 average drought conditions ($p_V$<0.25) according to (a)**
**monthly precipitation (P1), (b) simulated soil moisture (SM), (c) streamflow (Q), (d) groundwater (GW), or (e) all months in which at least one impact started.**

The medium agreement between various drought hazard information sources (Fig. 7a-d) and the fact that not one single drought information source fully agreed with the start of drought impacts (Fig. 7e) motivates a further exploration of the advantages of using local multivariate over single variable regional average drought information (Fig. 8). Characterizing the drought of 2018-

2019 by single variable NUTS-1 average drought information provides an incomplete picture and fails to predict part of the drought impacts (Fig. 8a,c). For example, NUTS-1 average P1 picked-up the peak of the drought in 2018 (Fig. 8a). However, NUTS-1 average P1 missed the prolonged duration of the NUTS-1 average GW drought in 2019 and associated impacts, i.e., NUTS-1 average P1 does indicate drought recovery in winter 2018-2019. In contrast, NUTS-1 average GW missed the occurrence of drought in the summer of 2018 and fails to predict the manifold of impacts that occurred. NUTS-1 average P1

also missed the occurrence of some NUTS-1 average SM and Q drought months and associated impacts in the summer of 2019. The difference between NUTS-1 average and local drought information revealed that average drought information




missed the early onset and delayed recovery of part of the local hydrological drought conditions. According to the regional average drought signal, the drought of 2018 started in April-May (Fig. 8a). However, local drought information revealed that drought conditions developed earlier for part of the study region, which matches with the earlier start of impacts related to

public water supply in winter as reported for some NUTS-2 regions (Fig. 8b,c). The same was observed for GW, i.e., the NUTS-1 average GW drought of 2018 started in autumn, whereas local GW conditions reached drought much earlier, which matches with the earlier occurrence of local summer impacts in 2018 that described wells running dry.

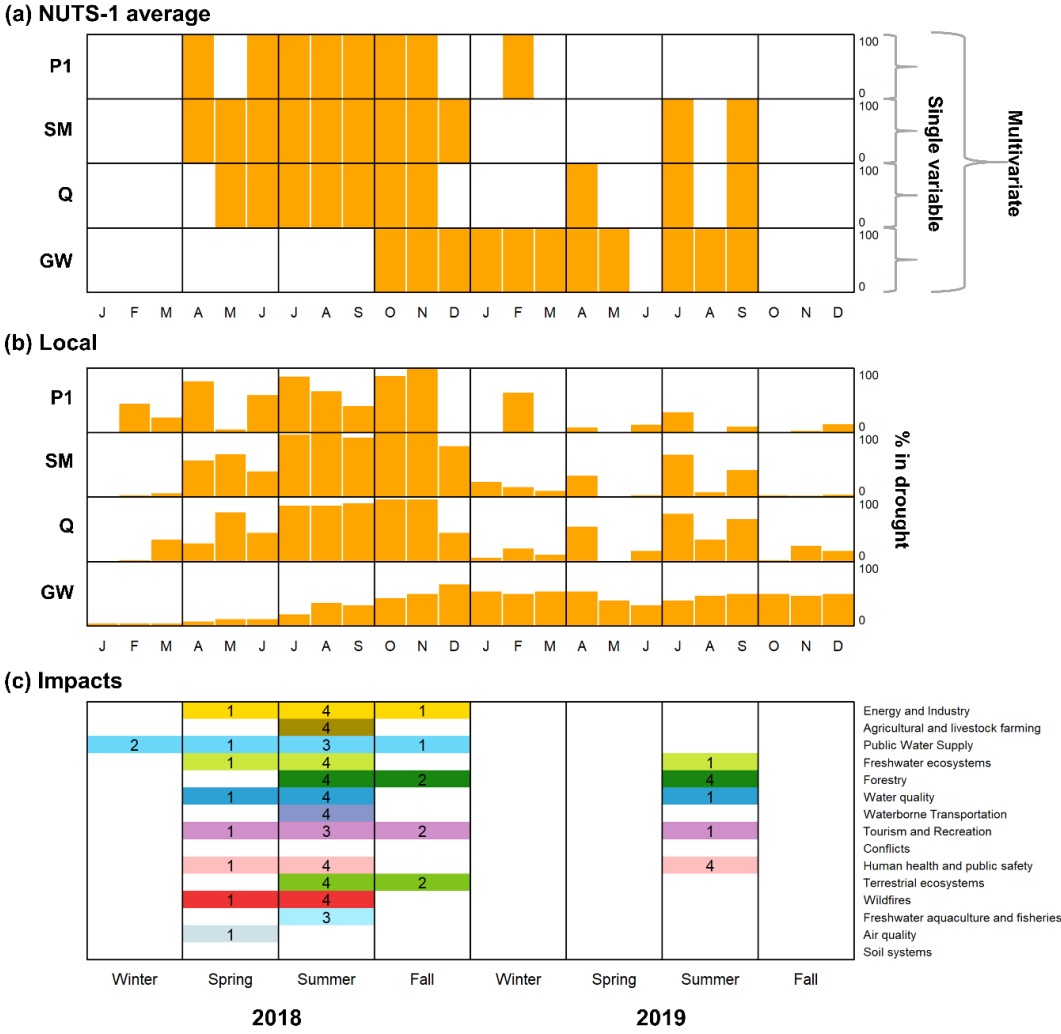

**Figure 8. Differences among drought information sources displayed for the drought of 2018-2019 ranging from regional average**
**single variable information to local multivariate drought information of precipitation (P1), simulated soil moisture (SM), streamflow (Q) and groundwater (GW): (a) NUTS-1 average and (b) local monthly drought hazard information expressed as the percentage of grid cells, catchments, or wells in drought. (c) Start of drought impacts by impact category and season, with numbers indicating for how many NUTS-2 regions impacts were reported (maximum 4).**



## 4 Discussion

### 4.1 Typical droughts and their hazard and impact characteristics


Our first objective was to identify and characterize past drought episodes in Baden-Württemberg based on a multi-variable assessment. The analyses revealed different types of drought, of which the intense multi-seasonal dry spells peaking in summer (Type I) that occurred in 2003, 2015 and 2018 were most prominent according to the duration and severity of the drought hazard and the amount and variety in drought related impacts (Fig. 2-4). The episodes of 2003 and 2015 are relatively well

known and documented (e.g., Ionita et al., 2017; Laaha et al., 2017; Van Lanen et al., 2016). Compared to these episodes, the drought of 2018 in the study area was generally more extreme in terms of the duration and severity of precipitation, soil moisture and river flow deficits, which is in line with findings for Northern Europe and Switzerland (resp. Bakke et al., 2020; Brunner et al., 2019). Our study further revealed a multi-year nature of the drought of 2018 that made the event more impactful, which is in line with findings for the groundwater drought in the Netherlands (Brakkee et al., 2021). Hydrological droughts

for part of the catchments and wells persisted far into 2019. This delayed development and prolonged recovery of hydrological drought has been reported for previous drought episodes in various studies (e.g., Parry et al., 2016; Peters et al., 2005).

The Type I drought episodes of 2003, 2015 and 2018 all coincided with some months in the growing season with above normal air temperatures. These above normal air temperatures revealed to have a compounding effect on drought impacts, contributed to soil moisture- or hydrological-drought development (e.g., Brunner et al., 2021) and could have caused a more rapid

intensification of drought conditions also known as "flash droughts" (e.g., Nguyen et al., 2019). The generally increasing air temperatures will exacerbate drought and its impacts and present a challenge for future drought management (Brunner et al., 2021; Markonis et al., 2021; Pendergrass et al., 2020). The type I events studied here might be seen as a precursor of intense warm-climate drought events and their typical impacts.

The impacts of Type I episodes in the study area were severe and diverse and thanks to the data collection in impact databases

such as the EDII well documented. Reported impacts in the sector "agriculture and livestock farming" were related to e.g., reduced crop yield and quality, a lack of food for livestock, or increased cost for irrigation water. Losses in agriculture are mostly attributed to low soil water availability and heat (see also e.g. Peichl et al., 2019), whereas the increased cost in irrigation relate to low (sub-)surface water levels and the consequent need to use alternative water sources. Impacts on forestry relate to tree growth and vitality and were associated with low soil moisture and groundwater levels in the impact reports, depending

on the kind and age of the tree (e.g. Skiadaresis et al., 2019). Energy and industry impacts relate to reduced hydropower production due to low surface water availability or the shutdown of hydropower plants for technical or ecological reasons, e.g., a lack of cooling water or the exceedance of ecological thresholds (see also e.g. Van Vliet et al., 2016). Furthermore, streams with a number of hydropower plants experienced hydropeaking and emerging water conflicts among users with different interests, e.g., energy production versus ecology (see also e.g. Bruder et al., 2016). Public water supply presented challenges

during drought, e.g. resulting in the use of alternative resources such as other wells or rivers. Nevertheless, water security was strong thanks to regional water exchange networks and long-distance water transfers (Blauhut et al., 2020). Merely few high-



altitude settlements had to be supported with deliveries. Impacts on aquatic and terrestrial ecosystems and ecology were also diverse, ranging from fish die-off to the spreading of plant diseases favored by drought conditions. We found that the drivers of these impacts were above normal temperatures (heat stress) and below normal water availability. Overall, the manifold of

causes of a large variety of drought impacts highlights the need for multivariate drought management.

Lesser known and studied are the other drought types. However, these drought types can also be associated with a variety of impacts and, therefore, deserve more attention and awareness. Type I→II droughts are mainly characterized by prolonged drought conditions in less responsive hydrological systems that develop after intense Type I droughts (Fig. 3, 5). Notable is their asynchronous development and recovery, i.e., hydrological drought can develop and recover long after meteorological

and soil moisture drought. This behavior might not be accurately represented in hydrological models, particularly some of the large-scale ones (Tallaksen and Stahl 2014). This asynchronous development meant that the slowly developing hydrological drought did not necessarily coincide with droughts in other domains of the hydrological cycle or with heatwaves. This meant that the impacts of Type I→II droughts were different compared to e.g. Type I droughts (Fig. 4). First, the number of reported impacts and categories is lower suggesting that Type I→II are less impactful and visible. Second, different impact categories

comprise larger shares in the overall impact category distribution. For example, impacts starting in the year 2019 mostly relate to forestry, as prolonged hydrological (groundwater) drought episodes might be especially critical for some tree species (Tegel et al. 2020). Furthermore, the effects of drought on trees can be creeping and accumulating; drought weakened trees can be affected by a variety of pests and diseases which can result in delayed diebacks (Schuldt et al. 2020). We further expect hydrologically induced impacts on e.g., public water supply that started during the Type I droughts to continue. However, this

needs to be further investigated, as most impacts only report a start and no end date, which aligns with the general challenge of identifying when a drought and its impacts are fully recovered (e.g. Parry et al., 2016). In addition, the survey to gather additional impacts on the public water supply sector for the impact database ended in 2018. Altogether, a relatively wet winter after a Type I drought episode might give a false sign of drought recovery, as some less visible hydrological deficits and consequent impacts linger on.

The Type II groundwater droughts of the early 1990s exceeds other groundwater droughts in both duration and severity (Fig. 3). These prolonged groundwater drought conditions might be associated with the long-term meteorological water deficits and the absence of distinct wet periods (Fig. 2), as also reported for parts of central Europe (e.g. Hannaford et al., 2011; Spinoni et al., 2015). Nevertheless, (changes in) groundwater abstraction cannot be ruled out for all wells and might have had an influence. Groundwater related impacts for this period were not available in the EDII for Baden-Württemberg. This absence of impact

reports in the EDII mostly relates to the lower research attention towards droughts in this period as well as to the lower impact data availability prior to the digital era. Nevertheless, neighboring NUTS-1 regions show evidence of impacts of this episode in the EDII related to e.g., cargo transport on the river Rhine or tree vitality.

Type III droughts describe shorter meteorological dry spells mainly affecting soil moisture and streamflow (Fig. 3). These meteorological dry spells were not intense enough to cause a strong decline in groundwater but could nevertheless coincide

with and worsen ongoing groundwater droughts (Type III+II). The Type III episode of 2011 and 2014 occurred before the





summer season. Their timing in occurrence, together with their relatively short duration, affected the impact categories that occurred, which were mostly related to energy (hydropower) and water use conflicts. Typical summer impacts enhanced by high air temperatures related to e.g., water quality or agriculture were absent. Remarkable about the drought of 2011 was the large share of impacts related to waterborne transportation, which related to low water levels in the Rhine towards the end of

spring but also in a very dry November (see also Kohn et al., 2014). The Type III+II episode of 1998 revealed the joint occurrence of soil moisture, streamflow, and groundwater drought. There is not much evidence in the EDII on the impacts of this episode given the general lower impact availability prior to the year 2000. Nevertheless, we hypothesize that such Type III+II episodes provide extra stress for e.g., irrigated agriculture or tree vitality, given the joint occurrence of limited soil moisture, streamflow, and groundwater supply.

Overall, different drought types share common characteristics and impacts. This similarity among drought types highlights the potential of drought type specific management and research (Markonis et al., 2021). Our results imply that such a typology should not only consider the timing of drought, but also the domain of the hydrological cycle in which the drought appeared, and whether droughts in different domains of the hydrological cycle as well as high temperatures coincided or not. Most well-known and impactful were the Type I drought episodes, and these episodes can be adopted as worst-case benchmarks for

drought management. However, the Type I drought episodes of the past decades never coincided with persistent long-term hydrological (groundwater) drought. An important question remains what would have happened when a drought episode like 2018 started when storages in hydrological systems were low, i.e., a Type I+II episode. Stress tests or scenario studies can be used to explore the impacts of such worst-case episodes as well as to prepare drought management for the future (e.g Grecksch, 2019; Hellwig et al., 2021; Stoelzle et al., 2020).

**4.2 Regional drought propagation characteristics**

Our second aim was to evaluate the (variability in) regional drought propagation signals. We showed that drought propagation of Type I droughts of 2003, 2015 and 2018 generally followed the hypothetical order and time lag of drought propagation as described in e.g., Van Loon (2015). This suggests that the concept of drought propagation could be a general guiding principle for regional drought management. However, we also found several deviations from the general concept of drought propagation

that need to be considered.

First, drought propagation does not have to complete the full cycle, which means that the expected ordering and lengthening is not always observed (Fig. 2, 5-6). For example, the relatively short meteorological droughts of 2011 and 2014 propagated to below normal soil moisture and river flow but groundwater levels often stayed in the normal range, especially for the less responsive aquifer types. On the other hand, short dry spells can severely threaten groundwater systems when the initial

conditions are low at the start of the dry spell as occurred in, e.g., summer 1998. This implies the importance of tracking non-drought conditions (approaching drought) in a drought management context, especially for the more slowly responding rivers and aquifers. For these slowly responding hydrological systems, the initial conditions can be an important precursor for the likelihood of future drought conditions (e.g. Parry et al., 2018). Another deviation from the general drought propagation





concept is the occasional absence of lengthening in propagating drought signals, which relates to the time-lag in the
development of deficits and consequent chances of having drought-recovering wet conditions prior to drought reaching its full
extent. For example, we showed for the prominent drought episode of 2015 that groundwater reached drought conditions after
a prolonged meteorological dry spell and associated persistent soil moisture and river flow droughts (Fig. 5-6). However, a
very wet period followed soon after the development of the still relatively small groundwater deficits, which meant a quick
recovery. Finally, the spatial variation in drought conditions revealed that local drought propagation over a larger region does
not necessarily follow the expected sequencing of different drought types (Fig. 6). For example, a responsive groundwater
aquifer with a low amount of storage reached below normal conditions before a less responsive river underlain by a high
storage aquifer (see also Stoelzle et al., 2014). The spatially variable response time was also visible in the impacts, e.g., the
start dates of local impacts on public water supply in 2018 ranged from early to late in the year (Fig. 8c).  These variable
hydrological responses to meteorological dry spells underpin the need for the consideration of local differences in response to
meteorological drought among different (hydrological) systems at different timescales (Vicente-Serrano et al. 2021; Wu et al.,
2021).

### 4.3 Disagreement among drought information sources

Our third aim was to assess the variability in drought hazard signals and consequent disagreement among drought information
sources. The found variability in propagating drought signals across variables and scales implies complexities in the use of
single variable or regional average (composite) drought information for comprehensive drought assessments, as relevant
variations are lost (Fig. 7-8). Single variable or regional average drought hazard information, especially of soil moisture and
streamflow, could predict the start of part of the drought impacts (Fig. 7e). However, regional average single variable drought
hazard information was not sufficient to predict the start of all drought impact occurrences. Part of the reason in the mismatch
between drought hazard information and the start of drought impacts might be related to inaccuracies in reporting. However,
another explanation might be the mismatch between drought hazard information sources, e.g., regional single variable
information might miss the earlier onset or delayed recovery of some (hydrological) systems  (Fig. 8). This suggests, together
with the variety of causes of drought impacts (Sect. 4.1), that the joint consideration of different variables at a local rather than
regional scale is needed to predict the full range of impacts during different stages of drought. Drought monitoring and
management would benefit from future efforts towards joint portals that include near real-time multivariate drought hazard
and impact information at different scales. Such portals might be used to design or implement operational definitions of
drought, paving the way towards more targeted, sector and location specific, drought management.

### 5 Conclusion

Past drought episodes and their impacts in southern Germany ranged from intense multi-seasonal drought episodes with severe
soil moisture and streamflow deficits and compounding heat waves to less intense but more prolonged multi-year drought





episodes and associated low groundwater levels. The identified drought types each share common hazard, propagation, and impact characteristics, which suggests that drought-type specific management options could be designed. Of the different drought types, intense multi-seasonal drought episodes with compounding heat waves (as in 2003, 2015 and 2018) caused the largest number and variety of impacts. This drought type might become a challenge for future drought management in a warming climate. The drought of 2018, as an example, stood out by the length of the meteorological dry spell and consequent

prolonged and severe soil moisture and river flow deficits. Groundwater deficits in 2018 generally developed slower but lasted well beyond the end of 2018. This multi-year nature of drought highlights the need of long-term drought management beyond the peak of the drought in summer.

Drought risk management components such as monitoring and early warning often rely on drought propagation assumptions and concepts such as sequencing, time lag and lengthening. Studying the regional propagation of droughts mostly confirmed

the existence of these concepts on a regional scale and can thereby guide regional drought management. However, we also found deviations from the general drought propagation concept as drought propagation did not always affect the full hydrological cycle and the length of the dry spell did not always increase. Further, the order of appearance and lengthening was not necessarily visible in local drought conditions e.g., streamflow drought in a less responsive river might develop later and persist longer compared to groundwater drought in a more responsive aquifer.

Regional average single variable drought hazard information favors easy interpretation. However, drought information derived from different variables at different scales can show a low agreement. Therefore, a drought assessment based on regional average single variable drought information can vary significantly from a multivariate assessment of local drought conditions and might consequently fail to predict the occurrence of some of the impacts. Overall, comprehensive drought management and assessments benefit from the appraisal of local multivariate drought signals.


**Competing interests.** The authors declare no competing interests.

**Data availabillity.** Data are available from sources mentioned in Section 2.2 or upon request from the providers cited in this Section.


**Acknowledgements.** This work contributes to the DRIeR project supported by the Wassernetzwerk Baden-Württemberg (Water Research Network), which is funded by the Ministerium für Wissenschaft, Forschung und Kunst Baden-Württemberg (Ministry of Science, Research, and the Arts of the State Baden-Württemberg) (grant no. AZ. 7532.21/2.1.6). We further acknowledge the DWD and LUBW for providing data. Financial support of the German Research Foundation (DFG) for

SDS@hd - Scientific Data Storage is acknowledged. All analyses were carried out with the open-source software R (https://www.r-project.org/).



**Author contributions.** Conceptualization (all), formal analyses and writing of first draft (ET), review and editing (VB, MS, LM & KS), visualization (ET, VB, MS), funding acquisition (LM, KS).

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
