# Peer review of "Different drought types and the spatial variability in their hazard, impact, and propagation characteristics"

_Natural Hazards and Earth System Sciences, 2021_

## Author Comment (AC1)

**Online reply to the comments of reviewer 1:**

*We appreciate the helpful comments (shown in black). We are confident that all of them can be easily addressed in a revision, should we be invited to submit a revised version. We briefly reply to them here online (in blue) for swift clarifications.*

(1) It is not clearly defined in the manuscript, whether the P & T & SM variables are averaged over individual catchments or average over the entire BW region is performed (e.g., in Figure 2). There is a large spatial variation in meteorological variables due to the Alps, so this needs clear clarification.

*In fact, there are differences among variables: only Q is averaged from gauging stations and hence catchments. For P, T & SM, three different spatial aggregations were considered. Local (individual grid cells) and regional averages over the individual NUTS-2 regions as well as over Baden-Württemberg (NUTS-1). For Fig. 2, we consider the NUTS-1 average drought signals to define the major drought episodes over the region. These episodes then serve as a starting point to further explore the spatial variation in local drought signals during major drought episodes later in the manuscript (Sect. 3.2 & Sect. 3.3). We will more carefully clarify this reasoning and the methods.*

(2) What is the soil moisture representation from the TRAIN model? Please, provide some more details about the representativeness of this model with respect to soil moisture observations.

*The model was recently used in a study on soil moisture drought (Tijdeman & Menzel, 2021), where a lot of detail is given, also about the representativeness of the model. We suggest that we can add some of that background here and will refer more clearly to previous model uses in Section 2.2.*

(3) The title of section 2.4 requires a more intuitive name.

*We agree. One suggestion might be a title more in line with the corresponding results section e.g.: "Different drought episode types and their characteristics and impacts". But we will revisit the title of Section 2.4 carefully in the revised manuscript.*

(4) Line 224: how do you distinguish between quick and slow developing drought? I have missed this definition.

*Quickly developing drought conditions refers to the moment of time when the first 5% of the local drought signals of a certain variable reached drought conditions ($5^{th}$ quantile of all local initiation times). Slowly developing drought conditions refers to the moment of time when 95% of the local drought signals reached drought ($95^{th}$ quantile of the local initiation time). We will clarify this in Section 2.5 of the manuscript.*

(5) Line 232-235: More clarification for these conditions is required, the current explanation is too brief.

*This comment is in line with the $3^{rd}$ comment of reviewer 2. We will provide more explanation and corresponding references on drought propagation processes and concepts in the introduction (Section 1). Then, in section 2.5, we will link this additional explanation to the hypotheses that are being tested, including those in lines 232-235.*

(6) Section 2.6, provide a formulation of A in mathematic form. Additionally, in analogy to line 248, explain the meaning of A=1.

*We will add a mathematical expression of A as well as the suggested explanation of the meaning of A=1.*

(7) Figure 2: Why P12 and T12 do not have not the same scale. P12 is monthly, T12 is annual. Would not it make more sense to have it the same?

*From a consistency standpoint this is a valid argument. However, we decided to have different timescales for P and T for the following reason. P12 is on a monthly moving temporal scale of 12 months to make it comparable to the commonly used SPI-12 (McKee et al., 1993). For temperature, it is not common to look at moving temporal scales but rather to fixed time periods (months, seasons, or years). To follow the more commonly used approach for temperature, we used a fixed monthly timescale to capture the more short-term anomalies as well as a fixed annual timescale to capture the general change in temperature that happened over the studied period. We will more carefully motivate this reasoning in section 2.2.*

(8) Ticks on the x-axis of fig.2: ticks should be displayed for 1.1.YYYY, rather than the current version.

*Thanks for pointing this out. The year labels were actually placed in the middle of each year (1.7.YYYY) but we realize, also from your next comment (9), that this is confusing and will make sure to have distinct axes labels in a revised manuscript.*

(9) Please, explain, what happened during the year 2005? There were also several months of exceptional drought conditions identified but never discussed.

*Thanks for pointing this out. These short-term drought conditions happened in the winter of 2005-2006 and are similar to the winter drought conditions experienced in the winter of 2016-2017 (also not discussed in the current version). We will discuss these short-term winter drought episodes, which did not have a range of impacts due to their timing, in a revised version of the manuscript.*

(10)  Regarding the impacts, considering just the number of reported impacts is a big simplification. Can you quantify them as well using more quantitatively (e.g., financial losses, crop-yield losses?)

*Unfortunately, this is not possible as rarely financial or loss-numbers were provided in the original text-based sources used. The EDII database therefore focuses on simple registration of occurrence with a qualitative description. The issue is discussed in detail in the references named.*

(11) Where is the statement on lines 423-424 supported by earlier presented results?

*We did only bring this up in the discussion and not in the results, as the impact data collection did not specifically focus on capturing the drivers of impacts. However, some impact reports also mentioned drivers which could be used for the interpretation of the results. We will more carefully state that this statement is derived from a more detailed interpretation of textual drought impact report descriptions.*

(12) Two sentences on lines 426-427 require reformulation.

*Thanks for pointing this out.*

(13) Discussion can be possibly extended with the following suitable references: https://doi.org/10.1088/1748-9326/aba4ca on impact assessment with text mining; https://doi.org/10.1088/1748-9326/abe828 on assessing multi-year droughts by different aggregation periods.

*We appreciate the suggestions of publications which overlapped with our submission. We would carefully consider them in a revised and extended discussion.*

(14) The current data availability statement is not sufficient. Please, provide your processed data presented in this manuscript on the online repository.

*We will put the data in an online repository in case of acceptance of the manuscript.*

References:
McKee, T. B., Doesken, N. J., and Kleist, J.: The relationship of drought frequency and duration to time scales. Paper presented at Proceedings of the 8th Conference on Applied Climatology, American Meteorological Society, Anaheim, USA, 17–22 January 1993, 1993.
Tijdeman, E. and Menzel, L.: The development and persistence of soil moisture stress during drought across southwestern Germany, Hydrol. Earth Syst. Sci., 25, 2009–2025, https://doi.org/10.5194/hess-25-2009-2021, 2021.

---

## Author Comment (AC2)

**Online reply to the comments of reviewer 2:**

*We appreciate the useful comments (shown in black) and are confident that we can address them in a revised version should we be invited to submit one. Here we reply briefly (in blue) how we might address the comments.*

One of the conclusions is that "multivariate" drought management is needed, in recognition of the complexity of the relationship between drought and drought impacts, especially at the local level. It may be more appropriate to talk about multi-sectoral drought management, recognizing that, for example, water suppliers, farmers and ecologists generally measure and manage drought and drought impacts within separate scopes of decision-making. A reference to whether or not cross-sectoral drought planning occurs and is coordinated by any central authority, or whether that question requires further investigation, would be helpful.

*This is a very good point though challenging to address globally. Our impression is that quite often drought management is indeed sectoral. In southern Germany there is still too little cooperation for example between agriculture and water management institutions with agriculture using weather forecasts and own crop models and surface water related water management using river flow monitoring. We suggest that for the revision we will search for some references and elaborate a bit more on the issue as suggested.*

It would be easier to read if the authors refrained from using variable abbreviations throughout the discussion, such as "Ipv" instead of "initiation time", "Q" for "catchment," etc. The statements on lines 230-231 and 234-235 would be easier to read if they were converted to sentences, because their current form requires readers to remember several different abbreviations.

*Agreed. We will avoid the use of variable abbreviations in the discussion and will also rewrite lines 230-231 and 234-235 to full sentences.*

Investigating the phenomena of "ordering, time-lag and lengthening" of drought at different spatial scales is central to this analysis. It would be helpful to provide more explanation and cite references. What does the conventional wisdom (the literature on ordering, time-lag and lengthening) say? Is that what is being tested in the hypotheses? Why or why not?

*This comment is in line with the 5ᵗʰ comment of reviewer 1. We will provide more explanation and corresponding references on drought propagation processes and concepts in the introduction (Section 1). Then, in section 2.5, we will link this additional explanation to the hypotheses that are being tested.*

Minor corrections:

*All minor technical corrections pointed out are highly appreciated and will be addressed in the revision.*

---

## Author Response (AR1)

Dear Editor,

Below, we present a list of changes made to the manuscript, which are implemented based on our online replies to the comments of Reviewers 1 and 2. The line and page numbers indicate where changes were made in the track-change version of the manuscript. For brevity, we do not repeat our full online replies to those comments, but here link the actual changes made in the revised manuscript to the numbered specific comments (#C) and provide a short description of the change made.

| Reviewer 1, Comments | | |
|---|---|---|
| # C | Change's location | Change made |
| 1 | Page 7, lines 197-200 | We clarified the reasoning behind averaging drought signals and our methodological approach. |
| 2 | Page 5, lines 133-136 | We added a brief description of the TRAIN model and references to studies showing the representativeness of this model, also for the considered study region. |
| 3 | Page 8, line 208 | We changed the title of Section 2.4 to a more intuitive name. |
| 4 | Page 9, line 241-243 | We clarified how we distinguish between quickly- and slowly developing droughts. |
| 5 | Page 3, line 65-74 | More background and references on drought propagation were added to the introduction. |
| 6 | Page 9-10, lines 262-285 | We provided a formulation A in math form (eq. 5-7) and added the requested analogous description what A=1 means. |
| 7 | Page 6, lines 153-155 | We added the explanation why we looked at a static annual (12 month) window for temperature. |
| 8 | Page 11, line 300-301 | We added a note to the caption of Figure 2 that the tick marks were placed on the first of July of the shown year. We prefer this option over the suggestion by the reviewer to at "01.01.YYYY", as the latter takes up much more room in the limited plotting space. |
| 9 | Page 20, line 505-507 | We discussed the short winter droughts of the winters of 2005-2006 and 2016-2017 |
| 10 | No changes | Quantification of impacts based on e.g. economic losses not possible with EDII data as discussed in the references mentioned in Section 2.2 (page 6, lines 162-173). |
| 11 | Page 18, line 453-454 | We added that statements that the type and cause of impacts were derived from textual impact descriptions in the EDII. |
| 12 | Page 19, lines 471-472 | We rephrased the two sentences. |
| 13 | Page 18, lines 441-442 | We added a reference to De Brito et al. (2020) to the discussion. |
| 14 | Page 22, lines 592-594 | We added all processed data (local percentile timeseries) of all drought hazard variables (P1, P3, P12, T1, T12, SM, Q, GW) to an online repository. We updated the data availability statement and added the DOI of the repository (10.5281/zenodo.6449934). |
| Reviewer 2, Comments | | |
| 1 | Page 1, line 12-26
Page 22, line 573-588 | We agree that multi-sectoral management is what is needed and meant to say that this will require a multi-variable monitoring and assessment beforehand to learn - among many other things like coordination of sectoral policies, conflict resolution mechanisms etc. (a topic beyond this study). We have no published reference to management |

| | | implementation in this region and also do not want to use references that break the reading flow in the conclusion and open up too much of a management topic, while our study is only analytical on drought variables and impacts. We therefore rephrased the conclusion (and the abstract) accordingly, removing reference to drought management, to focus it more on the main transferable conclusion. |
|---|---|---|
| 2 | Page 9, line 248-259 Page | We rewrote the hypotheses in Section 2.5 to full sentences. The previous hypotheses in formula form were kept but between brackets at the end of these descriptions. |
| 3 | Page 3, line 65-74 | More background and references on drought propagation were added to the introduction. |
| **R**eviewer 2, **M**inor **C**omments | | |
| 1 | Throughout the manuscript | We edited for English syntax throughout the manuscript (for example the mentioned sentence on line 92-93). If there are still syntax errors remaining, we are sure they will be picked up by the type-setting team of NHESS. |
| 2 | Throughout the manuscript. | We replaced "resp." with "respectively". |
| 3 | Page 5, line 129 | Rephrased the sentence |
| 4 | Page 15, line 382 | Added the missing word |